# A Patient-Derived Scaffold-Based 3D Culture Platform for Head and Neck Cancer: Preserving Tumor Heterogeneity for Personalized Drug Testing

**DOI:** 10.3390/cells14191543

**Published:** 2025-10-02

**Authors:** Alinda Anameriç, Emilia Reszczyńska, Tomasz Stankiewicz, Adrian Andrzejczak, Andrzej Stepulak, Matthias Nees

**Affiliations:** 1Department of Biochemistry and Molecular Biology, Medical University of Lublin, 20-093 Lublin, Poland; 62005@umlub.edu.pl (A.A.); emilia.reszczynska@mail.umcs.pl (E.R.); andrzej.stepulak@umlub.edu.pl (A.S.); 2Department of Plant Physiology and Biophysics, Institute of Biological Sciences, Faculty of Biology and Biotechnology, Maria Curie-Skłodowska University, Akademicka St. 19, 20-033 Lublin, Poland; 3Saint Jan of Dukla Oncology Centre of the Lublin Region, ul. Jaczewskiego 7, 20-090 Lublin, Poland; tstankiewicz@o2.pl; 4Department of Otolaryngology and Laryngeal Oncology, Medical University of Lublin, Jaczewskiego 8 Street, 20-954 Lublin, Poland; adrian.andrzejczak@umlub.edu.pl

**Keywords:** head and neck cancer (HNC), patient-derived cultures, tumor heterogeneity, 3D cell culture, cancer-associated fibroblasts (CAFs), partial epithelial-to-mesenchymal transition (pEMT), personalized medicine, tumoroids, hydrogel scaffolds, notch signaling

## Abstract

Head and neck cancer (HNC) is highly heterogeneous and difficult to treat, underscoring the need for rapid, patient-specific models. Standard three-dimensional (3D) cultures often lose stromal partners that influence therapy response. We developed a patient-derived system maintaining tumor cells, cancer-associated fibroblasts (CAFs), and cells undergoing partial epithelial–mesenchymal transition (pEMT) for drug sensitivity testing. Biopsies from four HNC patients were enzymatically dissociated. CAFs were directly cultured, and their conditioned medium (CAF-CM) was collected. Cryopreserved primary tumor cell suspensions were later revived, screened in five different growth media under 2D conditions, and the most heterogeneous cultures were re-embedded in 3D hydrogels with varied gel mixtures, media, and seeding geometries. Tumoroid morphology was quantified using a perimeter-based complexity index. Viability after treatment with cisplatin or Notch modulators (RIN-1, recombination signal-binding protein for immunoglobulin κ J region (RBPJ) inhibitor; FLI-06, inhibitor) was assessed by live imaging and the water-soluble tetrazolium-8 (WST-8) assay. Endothelial Cell Growth Medium 2 (ECM-2) medium alone produced compact CAF-free spheroids, whereas ECM-2 supplemented with CAF-CM generated invasive aggregates that deposited endogenous matrix. Matrigel with this medium and single-point seeding gave the highest complexity scores. Two of the three patient tumoroids were cisplatin-sensitive, and all showed significant growth inhibition with the FLI-06 Notch inhibitor, while the RBPJ inhibitor RIN-1 induced minimal change. The optimized scaffold retains tumor–stroma crosstalk and provides patient-specific drug response data within days after operation, supporting personalized treatment selection in HNC.

## 1. Introduction

Head and neck cancer (HNC) is the seventh most common cancer globally, resulting in 325,000 deaths per year. Tobacco, alcohol abuse, and oncogenic viruses, including human papillomavirus (HPV), are the main risk factors in HNC [1]. With an incidence of over 90%, the most common tumor type is head and neck squamous cell carcinoma (HNSCC), which originates from the keratinocytes of the mucosal epithelium in the oral cavity, pharynx, larynx, nasal cavity, and salivary glands. In contrast, adenoid cystic carcinoma (ACC) comprises only 1–2% of cases. ACC is typically a salivary gland malignancy, and its occurrence at the base of the tongue is considered rare [2]. The complex relationship between the squamous cancer cells, the tumor microenvironment (TME), and the extracellular matrix (ECM) provides a broad spectrum for potential therapeutic interventions, regardless of the clinical stage [3].

Cancer-associated fibroblasts (CAFs) are an essential component of the TME. The diversity of CAFs contributes significantly to the intra- and inter-tumor heterogeneity of HNSCC. Recent research has shown that CAFs comprise at least three or four functionally distinct subtypes: myofibroblastic CAFs (myCAFs), inflammatory CAFs (iCAFs), vascular CAFs (vCAFs), and antigen-presenting CAFs (apCAFs) [4]. The most frequent myCAFs can be further subdivided into C1-type myCAFs, characterized by low expression of alpha smooth muscle actin (ACTA2) and high bone morphogenetic protein 4 (BMP4), in contrast to C2-type myCAFs, which show the opposite pattern [5]. The intra- and inter-tumor heterogeneity of HNSCC also includes different epithelial phenotypes. One of these simultaneously displays both epithelial and mesenchymal traits, undergoing a phenotypic shift described as partial epithelial-to-mesenchymal transition (pEMT). Recently, four major epithelial clusters were identified in HNSCC: (1) Sox2-high/K14-low cells with high density, (2) K14-positive large cells lacking pEMT and stem cell markers, (3) a stem-cell-like cluster expressing K14, Sox2, and Bmi1, and (4) cells co-expressing K14, Slug, and Vimentin [6] which have likely undergone a partial EMT.

Cisplatin is a standard chemotherapeutic drug used in HNSCC treatment, while advanced or metastatic cases (RM-HNSCC) often receive combination therapy with docetaxel, cisplatin, and 5-fluorouracil. Targeted treatments include monoclonal antibodies like cetuximab and bevacizumab, and pathway-specific inhibitors such as temsirolimus and rapamycin targeting mTOR signaling [7]. There are also Notch signaling modulators under pre-clinical investigation, such as RIN-1, which can context-dependently modulate Notch signaling by inhibiting RBPJ, whereas FLI-06 acts as a potent Notch inhibitor. RIN-1 targets RBPJ, perturbing the NICD–RBPJ–MAML (Notch intracellular domain–RBPJ–Mastermind-like) activator complex and the RBPJ–SHARP (SMRT/HDAC1-associated repressor protein) repressor complex; as a result, it behaves as a context-dependent modulator of Notch output [8]. In contrast, FLI-06 inhibits the early secretory pathway (ER-to-Golgi trafficking), preventing proper Notch receptor maturation and functionally blocking Notch signaling [9].

Three-dimensional cell culture systems overcome the limitations of two-dimensional models by better replicating the morphological and functional features of the tumor microenvironment. Three-dimensional cultures can be based either on scaffold-free or scaffold-based systems. Scaffold-free systems promote the spontaneous formation of multicellular aggregates, also known as organoids, and are typically facilitated by non-adherent and suspension-culture techniques. In contrast, scaffold-based cultures normally rely on synthetic (ceramics, metals, or polymers) or natural scaffolds (polysaccharides, fibrous proteins, ECM-derived, or decellularized matrix) and most frequently include diverse hydrogels [10,11]. Type I collagen is one of the most widely used hydrogels in 3D cell culture [12]. Patient-derived organoids (PDOs) have emerged as promising platforms for cancer research, with recent studies demonstrating their physiological and biological relevance for personalized medicine in HNSCC [13,14]. However, current 3D organoid protocols face diverse challenges, including variable success rates for their establishment, slow growth, small volumes and cell numbers, and the characteristic loss of stromal components during culture [15].

Previously, we described a simplified 3D cell culture method that involves seeding both tumor and stromal cells on top of ready-made Matrigel/type I collagen gels, rather than embedding the cells inside the gel. Embedding the cells homogeneously inside the gel, to which we refer as the “sandwich model”, sufficiently supports the growth of organoids from established cancer cell lines and promotes tumor/stroma co-culture with CAFs [16]. However, the differential impact of these two approaches and their pros and cons on the composition of primary cultures of patient-derived tumor cell suspensions remains poorly investigated. In this study, different variations in seeding tumor cell suspensions were used and compared for patient-derived tumor cell cultures, such as seeding the cells on top of the gel either more widely dispersed versus centered on a single spot, and by embedding the cells inside the gel, either dispersed or centered on a single spot. This is meant to impact cell densities, dispersion, and provide differential access to oxygen. 

The main goals were as follows: (a) to provide a rapid adaptation of patient-derived cell suspensions to 3D cell culture conditions after enzymatic tissue digest, (b) to propagate these cells in simple 2D cultures with different growth media and to determine which media keep the original tumor heterogeneity best, and (c) to generate a simple, robust, reproducible, informative, and physiologically relevant 3D platform for in vitro drug sensitivity assays in personalized medicine.

## 2. Materials and Methods

### 2.1. Patient Information and Tumor Sample Preparation

Tumor specimens were obtained during routine surgeries performed by two independent surgical teams at the Oncology Center of the Lublin Region (Centrum Onkologii Ziemi Lubelskiej, COZL) and the University Clinical Hospital No. 4 in Lublin (Uniwersytecki Szpital Kliniczny Nr 4 w Lublinie), Poland. Preoperative diagnostic evaluation was performed through biopsy to identify the tumor type. Tumor biopsies were delivered to our laboratory on the same day as the surgery. Written informed consent was obtained in accordance with the local Ethics Committee guidelines (KE-0254/96/2020 and KB-0024/134/09/24), following institutional and national guidelines for the use of human material. Patient information is given in Table 1.

### 2.2. Preparation of Patient-Derived Suspension Cells and Isolation of CAFs

Tumor samples were washed with DPBS (Thermo Fisher Scientific, Lenaxa, KS, USA), minced with a scalpel into pieces < 1 mm, and dissociated using the gentleMACS™ Tissue Dissociator (Miltenyi Biotec, Bergisch Gladbach, Germany) for more efficient homogenization before enzymatic dissociation. The homogenized samples were digested in the first round with 0.25% Trypsin-EDTA (Sigma-Aldrich, St. Louis, MO, USA) for 30 min at 37 °C. The suspension was then filtered through a 100 µm cell strainer (Corning Falcon, Corning, NY, USA). The filtered fraction, which is enriched in single cells and small clusters, was centrifuged at 900 RPM for 5 min and cryopreserved in 90% heat-inactivated FBS and 10%DMSO (both Sigma-Aldrich) in liquid nitrogen for later culturing. The undigested tissue remaining on the 100 µm filter was subjected to a second round of digestion with 0.25% Trypsin-EDTA for 60 min at 37 °C. This material was then sequentially filtered through 100 µm, 70 µm, and 40 µm strainers and cells centrifuged at 1500 RPM for 10 min. The resulting single-cell suspension was cultured in T25 flasks with Advanced DMEM/F-12 (Thermo Fisher Scientific), supplemented with 1% penicillin-streptomycin (10,000 U/mL), Primocin (100 μg/mL, Invivogen, San Diego, CA, USA), and 18% FBS under 5% CO_2_ at 37 °C. This high-FBS condition was used to suppress the growth of squamous carcinoma cells, instead promoting the outgrowth of CAFs. After 2 days, nonadherent cells were removed, and fresh medium was added. CAFs were expanded until they reached 80–90% confluency, typically by passage 3, and subsequently frozen for later use.

### 2.3. IF Staining and Western Blot Analysis for Identification of CAF Subtypes and Preparation of CAF-Conditioned Medium

CAFs exhibiting distinct morphologies, as described in [5], were selected for IF staining using an anti-alpha smooth muscle actin (α-SMA) antibody (Abcam, Cambridge, UK, catalog # ab124964) and nuclear counterstaining with Hoechst 33342 (Thermo Fisher). The IF procedure was performed as previously described in [17]. Western blot analysis for CAFs was performed as previously described [18] with slight differences. Instead of 20 μg of total protein extract, 40 μg of protein extract was electrophoresed on 10% (*w*/*v*) and 5% (*w*/*v*) sodium dodecyl sulfate-polyacrylamide gel electrophoresis (SDS-PAGE) gels according to the size of the protein. The primary antibodies used were α-SMA (ab124964, Abcam, Cambridge, UK), anti-BMP4 (ab39973, Abcam), anti-Vimentin (sc-6260, Santa Cruz Biotechnology, Dallas, TX, USA), and anti-NOTCH3 (2889S, Cell Signaling Technology, Danvers, MA, USA). The secondary antibodies used were Horseradish peroxidase-conjugated (HRP) anti-rabbit IgG and anti-mouse IgG (both Cell Signaling Technology, Danvers, MA, USA, #7074P2 and #7076P2). Pierce ECL Western Blotting Substrate (Thermo Fisher Scientific) was used to detect and quantify proteins, and the results were visualized using a G:BOX Mini 9 Multi-Application Gel Imaging System (Labgene Scientific, Châtel-Saint-Denis, Switzerland). After the identification and characterization, myCAFs at passage four were cultured in T-75 cell culture flasks with DMEM/F-12 containing 10% FBS until they reached 80–90% confluence. CAF-conditioned media were collected, centrifuged at 900 RPM for 5 min at 4 °C to remove non-attached cells and cell debris, and then filtered through a Millex MCE syringe filter with a 0.22 μm pore size (Merck Millipore, Darmstadt, Germany). The medium conditioned by C2-type CAFs and C1-type CAFs was mixed in a 1:1 ratio and was kept at −80 °C until used in our tissue cultures.

### 2.4. Preparation of Hydrogels for Direct 3D Culture of Patient-Derived Tumor Cells

Matrigel (phenol-red-free, Matrigel; Corning Inc., Corning, NY, USA), type I collagen (rat tail collagen type I; Corning), HEPES buffer (1 M; Gibco, Thermo Fisher Scientific), and hyaluronic acid (HA) were used to prepare the MCH hydrogel. Hyaluronic acid was prepared as described previously with minor modifications [19]: Briefly, 50 mg of Hyaluronic acid sodium salt (from Streptococcus, Thermo Fisher Scientific) was stirred with 5 ml of PBS containing Ca^2+^ and Mg^2+^ ions overnight. The pH was measured and adjusted to neutral pH 7.0. Gels were finally prepared at a final concentration of 2 mg/mL Matrigel, 1 mg/mL type I collagen, and 2% HA (*w*/*v*). A total volume of 600 µL gel mix was used for each well of a 6-well plate (Corning ^®^ Costar ^®^ TC-Treated Multiple, Corning Inc., Corning, NY, USA) and allowed to polymerize for 2 h in an atmosphere of 5% CO_2_ at 37 °C. Tumor cell suspensions were unfrozen and seeded on top of the prepared gels, at a density of 500,000 cells/well in our “media mix” composed of 1/3rd of DMEM/F12 with 10% FBS, 1/3rd of CAF-conditioned medium, and 1/3rd of growth-factor-enriched Endothelial Cell Growth Medium 2 (ECM-2, PromoCell GmbH, Heidelberg, Germany). After 2 days, non-attached, dead, and dying cells, cell debris, or undigested tissue fragments remaining were removed, the gel was washed with PBS, new medium was added, and the cells were cultured for up to 14 days. The morphological changes in primary cultures were visualized using the EVOS Cell Imaging Systems (ThermoFisher) on days 2, 7, and 14.

### 2.5. Transfer of Primary Tumor Cultures from 3D to 2D Conditions by Gel Digestion

To facilitate and simplify patient-derived tumor cell culture, we aimed to establish a straightforward and standardized 2D expansion protocol, followed by monitoring phenotypic changes that spontaneously emerge under different cell culture media. After 14 days in 3D culture, the Matrigel/Collagen/Hyaluronic Acid (MCH) gel (final composition: 2 mg/mL Matrigel, 1 mg/mL collagen type I, and 2% HA *w*/*v*) was digested by gentle stirring with a solution of 3 mg/mL of Dispase II (Gibco™, Gibco Cell Culture Solutions|Thermo Fisher Scientific) at 37 °C for 1 h and subsequently centrifuged at 4 °C at 2000 RPM for 10 min. After the supernatant was removed, the pellet was washed with cold PBS and centrifuged at 4 °C at 1500 RPM for 5 min to remove any remaining gel fragments.

### 2.6. Expansion and Downstream Applications of 2D Cultures

After the isolation of cells from MCH gel, 100,000 cells/well were cultured directly on uncoated plastic surfaces using five different media: (1) high glucose DMEM supplemented with 2.5% FBS, (2) DMEM/F12 with 18% FBS; (3) “Media Mixture 1” (ECM-2, CAF-conditioned medium, DMEM/F12 with 10% FBS as described above at a ratio of 1:1:1), (4) undiluted ECM-2 endothelial cell media with 2% FBS, and (5) “Media Mix 2” (50% ECM-2, 50% CAF-conditioned medium at 1:1 ratio). Cells were passaged 3 times with the same medium. The entire process of cell culture adaptation from 3D to 2D is represented in Figure 1.

After three passages, primary cell cultures maintained in the five different media were divided into three groups and seeded in 6-well plates. The first group was further passaged for subsequent experiments. The second group (300,000 cells/well) was cultured until reaching 80–90% confluency, then fixed with 4% PFA for immunofluorescence staining. The third group (500,000 cells/well) was directly harvested for RNA extraction and quantitative real-time PCR (qRT-PCR).

### 2.7. Immunofluorescence (IF) Staining for 2D Cultures

IF staining was performed as previously described [17]. After final washing with PBS-BSA 3 times for 5 min, cells were imaged using a Nikon A1R-Si HD Confocal Microscope (Nikon Instruments Inc., Melville, NY, USA). For non-adherent spheroid-like aggregates formed in ECM-2 cultures and tumor-like structures arising during late-stage Media Mix 2 cultures, detached material was collected from the medium, centrifuged, and fixed in 4% PFA. These structures were subsequently processed and stained using an optical clearing protocol for 3D cultures [20], without the need for embedding or sectioning. The primary antibodies used were anti-E-Cadherin (ab76055, Abcam), anti-Vimentin (ab45939, Abcam), Anti-pan Cytokeratin (ab7753, Abcam), anti-beta-catenin (MA1-301, Invitrogen Inc., Carlsbad, CA, USA), and anti-alpha smooth muscle actin (α-SMA) antibody (ab124964, Abcam). Goat anti-Mouse IgG (H+L) Cross-Adsorbed Secondary Antibody, conjugated with Alexa Fluor™ 647 (A-21235, Thermo Fisher Scientific Inc.), and Goat anti-Rabbit IgG (H+L) Cross-Adsorbed Secondary Antibody, conjugated with Alexa Fluor™ 555 (A-21428, Invitrogen) were used as secondary antibodies. Hoechst 33,342 dye was used for nuclear counterstaining.

### 2.8. Quantitative Real-Time PCR (qRT-PCR) to Monitor Changes in mRNA Gene Expression in 2D Cultures Treated with Different Media Compositions and Serum Concentrations

Total RNA was isolated using the RNeasy Mini Kit (QIAGEN GmbH, Hilden, Germany). The High-Capacity cDNA Reverse Transcription Kit (Applied Biosystems, Foster City, CA, USA) was used for the reverse transcription of 1 µg of total RNA. The SYBR Premix Ex Taq reagent (TaKaRa, Dalian, China) was used for qPCR analysis on an ABI PRISM 7500 real-time PCR system (Applied Biosystems), employing SYBR Green Universal Master Mix (Applied Biosystems) for qPCR. Relative mRNA expression was calculated using the 2^−ΔΔCt method. For each sample, target gene Ct values were normalized to GAPDH as the internal control (ΔCt). The ΔΔCt values were then determined relative to Media Mixture 1, which served as the calibrator (set to 1.0 for each gene). Fold change was expressed as 2^−ΔΔCt.

Specific primers were used as follows:GAPDH Forward: 5′-GAACGGATTTGGCCGTATTG-3′, Reverse: 5′-TTTGGCTCCACCCTTCAAG-3′;CTNNB1 Forward: 5′- ATCCAAAGAGTAGCTGCAGG-3′, Reverse: 5′-TCATCCTGGCGATATCCAAG-3′;PDGFR-β Forward: 5′-AGGACAACCGTACCTTGGGTGACT-3′, Reverse: 5′-CAGTTCTGACACGTACCGGGTCTC-3′;VIM Forward: 5′-AGGAAATGGCTCGTCACCTTCGTGAATA-3′; Reverse: 5′-GGAGTGTCGGTTGTTAAGAACTAGAGCT-3′;SNAI2 Forward: 5-ATCTGCGGCAAGGCGTTTTCCA-3′, Reverse: 5′-GAGCCCTCAGATTTGACCTGTC-3′;SOX2 Forward: 5′-TACAGCATGTCCTACTCGCAG-3′, Reverse: 5′-GAGGAAGAGGTAACCACAGGG-3′;TNC Forward: 5′-GGTACAGTGGGACAGCAGGTG-3′, Reverse: 5′-AACTG GATTGAGTGTTCGTGG-3′.

### 2.9. Scaffold-Based 3D Cultures in µ-Plate 96-Well Plate

At passage 4, 2D-expanded cells were transferred again into 3D conditions to investigate the spontaneous differentiation and formation of tissue-like structures. For this purpose, we were using µ-Plate 96-well (Ibidi GmbH, Munich, Germany). Tumoroids’ morphology was evaluated based on three independent variables: The first factor was (1) gel composition, the second factor was (2) media composition, and the third factor was (3) differential oxygen availability and nutrient diffusion.

To evaluate the effects of different gel compositions, the 3D sandwich/single-point method was chosen initially for patient 1. The settings were prepared by first placing 30 µL of the bottom gel into each well, allowing the gel to polymerize for 2 h, followed by the addition of 10 µL of unpolymerized gel on top. Before polymerization, 5000 cells were injected with 5 µL of media into the center of the upper gel, where they formed a dense cell cluster. Four gel compositions were tested in DMEM/F12 with 10% FBS: (a) 4 mg/mL pure Matrigel, (b) 4 mg/mL Matrigel + 0.375 mg/mL collagen I, (c) 2 mg/mL Matrigel + 0.75 mg/mL collagen I, and (d) 2 mg/mL Matrigel + 1 mg/mL collagen INext, we aimed to investigate whether different culture media could allow us to recapitulate the invasive features observed with tumoroids in 3D cultures. Here, we aimed to prevent undesired matrix degradation and contraction, combined with the adhesion of hyperactive CAFs at the bottom of the plates. Since this was preferentially observed with collagen or Matrigel/collagen type I mixed gels, we decided to switch to pure Matrigel for this purpose. Accordingly, cells from patient one were cultured using the 3D sandwich/single-point cell seeding method in 4 mg/mL pure Matrigel, testing the following media conditions: (a) DMEM High Glucose with 2.5%FBS, (b) DMEM/F12 with 10% FBS, (c) complete Endothelial Cell Growth Medium 2 (ECM-2), and (d) Media Mix 2 (a 1:1 combination of full ECM-2 and CAFs-conditioned DMEM/F12 with 10% FBS). The IF staining was performed as described previously [16]. The IF staining was performed as described previously [16].After selecting 4 mg/mL pure Matrigel with Media Mix 2 as optimal conditions for tissue-like structures to emerge, patients 2 and 3 were assessed (in addition to patient one cells), using the same 3D cell culture conditions. Additionally, we aimed to address the impact of different cell seeding methods, which affect oxygen and nutrient diffusion, on tumoroid formation. Cell suspensions, transferred from 2D interim cultures of patients 1–3, were analyzed for this purpose (summarized and described in Figure 2). The morphology of tumoroids was evaluated using the complexity parameter, calculated as: Complexity = perimeter^2^/(4π × area) (adapted from [21]). ImageJ software (version 1.54p) was used for perimeter and area measurements [22], with a pixel-to-micrometer ratio of 0.80 pixels/μm.

### 2.10. Drug Testing in 3D—Phenotypic Assays Combined with the WST-8 Metabolic Assay

For in vitro drug sensitivity testing in 3D tissue-like conditions (or as “tumoroids”), 5000 cells per well from patients 1–3 were grown in Media Mix 2 (full ECM-2 and CAF-conditioned medium mixed at a 1:1 ratio). Cell suspensions were trypsinized and cultured in single wells of the 96-well µ-angiogenesis plates (Ibidi GmbH, Munich) by directly seeding the cells on top of a 4 mg/mL Matrigel layer (protocol c), as shown in Figure 2, and were cultured for 5 days. Under these conditions, cell cultures rapidly formed large, complex three-dimensional multicellular aggregates, which rapidly developed strong aggressive/invasive properties, indicating high levels of cell motility. After 5 days, these tissue-like 3D aggregates were treated with 5 µM of cisplatin, 5 µM of the context-dependent Notch pathway modulator RIN-1, and 5 µM of the Notch pathway inhibitor FLI-06 for 2 days. The morphological changes in the complex were monitored and documented by phase-contrast microscopy using the EVOS Cell Imaging System (Thermo Fisher Scientific, USA). Changes in aggregate size (total area change by the drug were morphometrically measured and quantified by using Image J software (version 154, released 8 November 2023). Cell Counting Kit 8 (WST-8/CCK8, ab228554, Abcam) was used to determine the number of living cells by Multimode microplate reader Infinite M200 PRO (Tecan, Männedorf, Switzerland).

### 2.11. Statistical Analysis

All statistical analyses were conducted with GraphPad Prism (version 1.4.1, GraphPad Software, Boston, MA, USA). For monitoring changes in mRNA gene expression as a result of 2D and 3D cultures in different media conditions, two-way ANOVA was used. To evaluate the changes in complexity for the different cell seeding methods, repeated measures by one-way ANOVA were used. For tumoroid area measurements and WST-8 results, a two-way ANOVA with multiple comparisons using a post hoc test was employed. All data are presented in the form of mean ± SD and are the result of no less than 3 independently performed experiments. Statistical significance was determined as values of * *p* < 0.05, ** *p* < 0.01, *** *p* < 0.001, **** *p* < 0.0001, and ns: not statistically significant.

## 3. Results

### 3.1. Identification of C1 and High α-SMA C2 Subtypes of CAFs by IF Staining and Western Blot

Out of four patient-derived CAF cultures, two representative subtypes were selected based on cellular morphology observed by light microscopy and proliferation [5]. The rapidly proliferating, less elongated C2-type CAFs exhibited strong α-smooth muscle actin (α-SMA) staining, whereas the elongated, slower-growing C1-type CAFs showed weaker α-SMA signals and lower cell density. Western blotting confirmed subtype-specific protein expression. C2-type CAFs exhibited higher levels of α-SMA and NOTCH3, but lower levels of BMP4, in contrast to the C1-type CAFs, which showed lower levels of α-SMA and NOTCH3 but higher BMP4 expression. Vimentin expression was comparable between the two subtypes. Representative IF and Western blot images are shown in Figure 3, with complete uncropped blots provided in Appendix A.

### 3.2. Culturing of Patient-Derived Tumor Cell Isolates on Top of Matrigel/Type I Collagen/Hyaluronic Acid (MCH) Gel

Figure 4 illustrates the adaptation process used for cell suspensions isolated from patient biopsies on top of the MCH gel over 2, 7, and 14 days. By day 2, all three patients formed small tumor cell aggregates, which differed in density and organization. Since CAFs isolated from Patient 4 displayed a C2-type and strong hyperproliferative phenotype, these cells were used only for CAF medium preparation and stromal characterization (Figure 3). All subsequent drug-sensitivity testing experiments were conducted with Patients 1–3 only to maintain consistency across assays. In patient 1, the cells formed dense, string-like arrangements, which were later followed by steady expansion across the gel surface over time. Patient 2 exhibited the earliest signs of invasive behavior, with irregular aggregates showing outward cellular migration as early as day 2. These structures expanded and dispersed over time, forming multiple aggregates with a migratory pattern throughout the gel. Patient 3, derived from adenoid cystic carcinoma, exhibited aggregate fusion and compaction over time without extensive invasion into the surrounding matrix; they remained more localized, forming large, tumor-like nodules rather than scattering throughout the gel.

### 3.3. Variation in Cell Culture Media Leads to Differential Cellular Composition in 2D Tissue-like Cultures

Media conditions strongly influenced the selection and retention of different cell types in 2D cultures, as demonstrated by strikingly different formation of multicellular structures and expression patterns of epithelial versus mesenchymal markers. This is summarized in Figure 5A. Low-FBS DMEM media favored cell survival of tumor and pEMT cells, while high-FBS DMEM supported the expansion of CAFs. Media Mixture 1 (which retained 33% growth-factor-enriched ECM-2) effectively promoted the spontaneous formation of squamous carcinoma-like morphology, characterized by tumor island–like structures surrounded by stromal zones. ECM-2 medium alone generated non-adherent spheroid-like aggregates consisting of tumor and pEMT cells but failed to retain CAFs, as verified by immunostaining of different aggregates for stromal and epithelial markers (Appendix A). In contrast, Media Mix2 (which retained 50% complete ECM-2 media and growth factors) fully supported the spontaneous aggregation of invasive spheroid-like aggregates (Figure 5B, top panel) that retained all cell types, including epithelial, pEMT, and CAF populations. These rapidly growing mixed 2D cultures eventually contracted and spontaneously formed thick, white/opaque, macroscopic tissue-like masses reminiscent of fibrosis (Figure 5B, bottom panel), and completely detached from the plastic plates. Cells in these contracted and dense tissue-like masses retained viability, but growth was slowed down.

### 3.4. qPCR Analysis Reveals Different Cell Types and Differentiation Stages Resulting from 2D Tissue-like Cultures Treated with Different Media Compositions

Primary tumor cell suspensions cultured in high glucose/low serum DMEM (2.5% FBS) expressed increased levels of beta-catenin (CTNNB1), platelet-derived growth factor receptor beta (PDGFRβ), and SRY-box 2 (SOX2) mRNA across all samples. Vimentin (VIM) expression varied depending on the culture, showing both increases and decreases. Cell cultures grown in full ECM-2 media further upregulated PDGFRβ, SOX2, Snail Family Transcriptional Repressor 2 or Slug (SNAI2), and tenascin C (TN-C), while generally reducing VIM expression. Media Mix 2 induced moderate increases in SNAI2, SOX2, and TN-C, with relatively stable VIM expression. Gene expression pattern of epithelial vs. mesenchymal markers in cell suspensions cultured in Media Mix 2 closely resembled those grown in Media Mixture 1, with CTNNB1 displaying variable regulation between cultures. Relative gene expression for patient cell cultures in 2D, exposed to different media as described in Figure 6.

### 3.5. Different Growth Conditions Result in Variable Morphologies in Scaffold-Based 3D Cultures

After selecting cells expanded in Media Mix as the standard source of tumor cells for further experiments, we started optimizing the 3D drug testing platform used for generating tissue-like tumoroids in 3D, investigating the impact of matrix, media composition, and topology of cell seeding.

To evaluate the impact of different hydrogel compositions on tumoroids’ morphology, cells were cultured in Matrigel with gradually increasing concentrations of collagen type I (Figure 7A), using DMEM/F12 (10% FBS) as standard media, and the 3D sandwich/single-point method as the cell seeding method of choice This strategy offers relative moderate nutrient conditions (with low levels of growth factors) and moderate oxygen availability, due to slow diffusion into the gels. The use of pure Matrigel as a matrix results in the formation of small, non-invasive tumoroids, characterized by a central area composed exclusively of squamous tumor cells, surrounded by a small number of fibroblasts with very low growth and motile activity. The incremental rise in type I collagen concentrations as ECM resulted in strikingly different morphologies of the tumoroids and functional activation of CAFs. Incremental addition of collagen eventually resulted in the rapid contraction of the entire 3D culture, followed by destruction of gel integrity after typically 5 days of culture. This was observed for all patients. The most stable 3D culture condition, which excluded gel contraction, was observed with pure Matrigel. However, we also observed that Matrigel did not support cell motility or invasiveness, and the resulting organoid-like structures were not representative of the original tumor histology and composition.

A.Next, we questioned if more invasive tumor/stroma co-cultures and multicellular, heterogeneous tissue-like tumoroids in Matrigel could also be obtained by switching between different media preparations (Figure 7B), with more or less serum and growth factor supplements, or by adding fibroblast-conditioned media as a potent support for cell growth and spontaneous formation of tissue-like structures. Cell seeding was performed as described for the previous set of experiments using the 3D sandwich model and the single-point cell seeding method, which yields high local cell densities while also strongly supporting cell survival and growth. High glucose DMEM media with low 2.5% FBS resulted in the formation of tumoroids that were primarily or predominantly composed of tumor cells. Still, it does not support the growth of CAFs. In contrast, both ECM-2 media and Media Mix 2 (CAFs-conditioned media and ECM-2 1:1 mix), which are both enriched in growth factors such as EGF, IGF1, FGF2, and VEGF, resulted in the rapid formation of large and heterogeneous tumoroids with strikingly invasive structures, with a significant contribution of highly active CAFs in these sustained tumor/stroma cocultures. These structures were similar to those formed in hydrogels with a high type I collagen composition, but did not require collagen for their formation.B.Lastly, we assessed the effects of oxygen and nutrient availability on each patient’s culture. For patients 2 and 3, the 3D sandwich/single-point method reproduced the most complex tumoroid morphology previously seen in patient 1. In contrast, the 3D sandwich/dispersed method resulted only in small, poorly developed, and slowly growing organoids, while the 3D on-top/single-point method produced the most complex, fast-growing, and irregular tumoroids, closely resembling native tumor architecture (Figure 7C). In addition to nutrient and oxygen gradients, the higher local cell density in single-point cultures may also increase cell–cell contacts, which could contribute to the more complex tumoroid phenotypes observed.

### 3.6. In Vitro Drug Sensitivity Testing on 3D Cultures with Patient-Derived Tumoroids

After evaluating Matrigel alone and in combination with type I collagen, (1) Matrigel was selected as the hydrogel for subsequent experiments, (2) Media Mix 2 as the most reproducible and effective media composition, and (3) the 3D on-top/single-point method as the most effective seeding method. Using these three essential conditions, tumoroids derived from patients 1, 2, and 3 were exposed to different small molecular weight inhibitors or drugs. Tumoroid morphology was monitored every day for 7 days (Figure 8). We measured the total area of the tumoroids as an indicator for both tumor growth and invasion, and the readout of the metabolic WST-8 assay as a measure for cell viability. Notch pathway inhibitor FLI-06 resulted in the shrinking of the total area of tumoroids and concomitantly decreased cell viability for each patient. Although brightfield images of FLI-06–treated cultures sometimes showed a diffuse “halo” around tumoroids, this represented shadows of detached/dead cells and was not included in quantitative area analysis, which focused only on compact viable structures. Regarding the response to cisplatin, patients 1 and 3 were clearly sensitive to the drug, while patient 2 was resistant. This differential cisplatin sensitivity likely reflects the clinical heterogeneity observed in patients, as approximately 30–40% of HNSCC patients exhibit primary cisplatin resistance [1]. Context-dependent Notch pathway modulator RIN-1 did not show any robust, reproducible effect on cell viability across patients. However, the Notch pathway modulator RIN-1 notably restricted the invasive behavior of the cisplatin-resistant tumoroids derived from patient 2 (but not from patients 1 or 3). 

## 4. Discussion

Only ~4% of head and neck cancer studies use 3D in vitro models, highlighting the need for more physiologically relevant systems [23]. PDXs remain the benchmark for drug testing but require months to establish and lose human stroma [24], while PDOs grow faster but often eliminate stromal components [25]. Our approach addresses these limitations by establishing patient-derived tumoroids that retain both CAFs and pEMT cells, enabling drug testing within just days after surgery.

Three parameters were critical for maintaining heterogeneity. First, scaffold composition: Matrigel supports epithelial growth via PI3K/Akt signaling [25,26,27,28] but fails to sustain CAFs. Adding type I collagen enhanced both tumor and stromal survival, while hyaluronic acid promoted proliferation and invasiveness through RHAMM signaling [29]. Second, media composition: standard organoid media are optimized for tumor cells but typically exclude stromal partners [25]. Our Media Mixture 1 combined CAF-conditioned medium, ECM-2, and DMEM/F12 with 10% FBS. ECM-2, although serum-reduced, contains potent growth factors (EGF, FGF5, IGF1, VEGF) [30,31]; CAF-conditioned medium delivers paracrine support for epithelial and mesenchymal cells [32]; and FBS supplies undefined but essential survival factors [33]. Together, these preserved tumor–stroma heterogeneity more effectively than standard media. Third, seeding topology: on-top/single-point seeding promoted oxygenation, nutrient access, and cell–cell contacts, leading to complex tumoroids, whereas dispersed/embedded methods restricted diffusion and left cells isolated, impairing growth [34].

Following the initial 3D adaptation on MCH gels, expansion in 2D enabled sufficient cell numbers for downstream assays while preserving tumor heterogeneity, including CAFs, which are key regulators of tumor progression and epigenetic modulation [35,36]. Five different media conditions or mixes were tested for their ability to support epithelial tumor cells, CAFs, and partial EMT or pEMT cells, identified by IF staining for E-cadherin (CDH1), vimentin (VIM), and their co-expression [37,38]. High-FBS DMEM favored CAFs but induced tumor cell differentiation [16], whereas low-FBS high-glucose DMEM supported tumor and pEMT-like populations [39,40]. Media Mixture 1, with intermediate FBS and reduced growth factors, preserved both tumor and stromal compartments and promoted spontaneous organization into “tumor islands,” closely resembling squamous carcinoma histology [41]. In contrast, full ECM-2 medium promoted epithelial/pEMT cells but excluded CAFs, consistent with its growth factor–driven bias toward epithelial proliferation [30,31]. Media Mix 2, which combined ECM-2 with CAF-conditioned medium, overcame this limitation by sustaining epithelial, CAF, and pEMT populations in parallel. Significantly, it also promoted ECM deposition and the development of invasive, tissue-like structures, thereby providing a more physiologically relevant model than either ECM-2 or Mix 1 alone.

These properties and the cellular composition retained in different culture conditions were further investigated by qRT-PCR for epithelial (CTNNB1), mesenchymal (VIM), stemness (SOX2, PDGFRβ), and pEMT-related (SNAI2, TN-C) markers [6,42,43]. High-glucose DMEM upregulated CTNNB1 across all patients, likely reflecting stress from low FBS conditions [44]. The mesenchymal marker VIM was reduced in ECM-2, suggesting CAF exclusion, while Media Mix 2 preserved VIM expression by maintaining stromal populations. Stemness markers SOX2 and PDGFRβ were elevated in both DMEM and ECM-2; although PDGFRβ can reflect mesenchymal features, its increase here more likely indicates enhanced tumor plasticity and stemness [43]. pEMT markers SNAI2 and TN-C were strongly induced by ECM-2 but balanced in Media Mix 2, which uniquely sustained epithelial, CAF, and pEMT compartments in parallel. Collectively, these data confirm Media Mix 2 preserved tumor heterogeneity while minimizing lineage drift, making it the most suitable platform for downstream 3D assays.

We optimized 3D growth conditions for chemosensitivity testing by developing a rapid, reproducible model that preserved tumor heterogeneity and invasiveness, offering a practical alternative to PDX models. Adding type I collagen enhanced tumor–CAF crosstalk but caused gel contraction and failure due to CAF hyperactivation via α2β1/α11β1 integrins and FAK signaling [45]. To maintain stability, Matrigel was selected as the scaffold, and different media compositions were tested to ensure CAF support; among these, Media Mix 2 proved most effective in sustaining invasive, heterogeneous tumoroids. Among three seeding strategies, the on-top/single-point method produced the most complex, irregular tumoroids, likely due to improved oxygenation, nutrient diffusion, and increased cell–cell interactions. These morphological differences were quantified using the complexity parameter (perimeter^2^/4π × area), which captures shape irregularity and heterogeneity [21,46]. By contrast, dispersed embedding generated uniform but poorly proliferating structures with limited complexity and weak CAF support, likely due to restricted diffusion of soluble factors and oxygen [47,48].

Finally, we applied the optimized tumoroid system to chemosensitivity testing with cisplatin and Notch modulators. The Notch inhibitor FLI-06 [9] was the most effective, consistently reducing viability across all three patient-derived tumoroids, likely through induction of apoptosis and G0/G1 arrest, consistent with prior findings in other solid tumors [49] and HNSCC models [9]. Cisplatin showed heterogeneous efficacy: two cultures were sensitive, while Patient 2 was resistant, reflecting clinical variability. The Notch modulator RIN-1 [8] had little impact on Patients 1 and 3. Still, it selectively limited the growth of Patient 2’s cisplatin-resistant tumoroids. As shown in the clinics and various models, increased NOTCH1 signaling/activity often correlates with reduced cisplatin sensitivity. Experimental Notch blockade has been demonstrated to re-sensitize cells to cisplatin. Mechanistically, Notch sustains cancer stem-like populations and EMT/invasion programs (e.g., via the NOTCH1/NOTCH3–JAG1 axes), while also rewiring the DNA-damage response and pro-survival signaling pathways (PI3K/AKT, STAT3, NF-κB), thereby blunting cisplatin-induced apoptosis [50]. In this framework, our finding that the NOTCH modulator RIN-1 selectively restricted radial spread/area in the cisplatin-resistant tumoroids without loss of viability is consistent with other reports showing that Notch perturbation enforced a cytostatic/anti-invasive state, a phenotype also seen when Notch output is modulated in other squamous models. Prior studies have demonstrated that RIN-1 targets RBPJ and induces activation-like increases in Notch-responsive transcripts (e.g., HES1/HEY1, NOTCH3, JAG1) in specific HNSCC settings [8,51]. Furthermore, disrupting RBPJ can de-repress specific Notch targets or indirectly enhance Notch tone via ligand changes, resulting in a cytostatic, differentiation-inducing program (increased expression of HES/HEY-linked p21 CDKN1A, reduced proliferation and invasion) rather than promoting apoptosis. We propose that cisplatin-resistant tumoroids retain a Notch-responsive, stem cell- and/or EMT–enriched state. We observe that RIN-1 likely perturbs this circuitry, thus effectively reducing radial spread without an apparent loss of viability.

## 5. Study Limitations and Future Directions

Our study’s primary limitation is the small patient cohort (n = 4, with n = 3 for drug testing), which limits statistical power for biomarker discovery. However, our findings provide necessary proof of concept for the platform’s feasibility and demonstrate patient-specific drug responses that warrant validation in larger cohorts. Additionally, we did not obtain treatment outcome data for clinical correlation analysis, which represents an important future direction. Recent clinical trials, such as the SOTO study, are investigating organoid-guided treatment selection in HNSCC patients and may provide frameworks for future validation studies [52]. Another significant limitation of our current system is the absence of immune cells, which are central components of the tumor microenvironment (TME). Immune cells, including tumor-associated macrophages, T cells, and myeloid-derived suppressor cells, play a critical role in influencing tumor progression and therapeutic response [53]. Their integration into 3D culture systems has the potential to enhance physiological relevance and allow investigation of immunomodulatory therapies. Future studies could address this by incorporating autologous immune cells or peripheral blood–derived immune populations into the co-culture system. Other limitations are the absence of histopathological sectioning and direct comparison of tumoroids with matched patient tumors. Future studies will incorporate histological evaluation by pathologists for further validation.

## 6. Conclusions

We developed a robust and reproducible scaffold-based 3D culture platform for personalized medicine in head and neck cancer (HNC). Starting from fresh patient tumor biopsies, our system enables rapid in vitro adaptation, expansion, and reconstruction of tumor microtissues with persistent heterogeneity, including stromal and pEMT components. Through the optimization of hydrogel composition, media, and oxygen exposure, we established a functional, patient-specific drug testing system. This clinically connected platform bridges surgical oncology and laboratory research, providing a practical and time-efficient tool for guiding individualized therapeutic decisions and personalized medicine.

## Figures and Tables

**Figure 1 cells-14-01543-f001:**
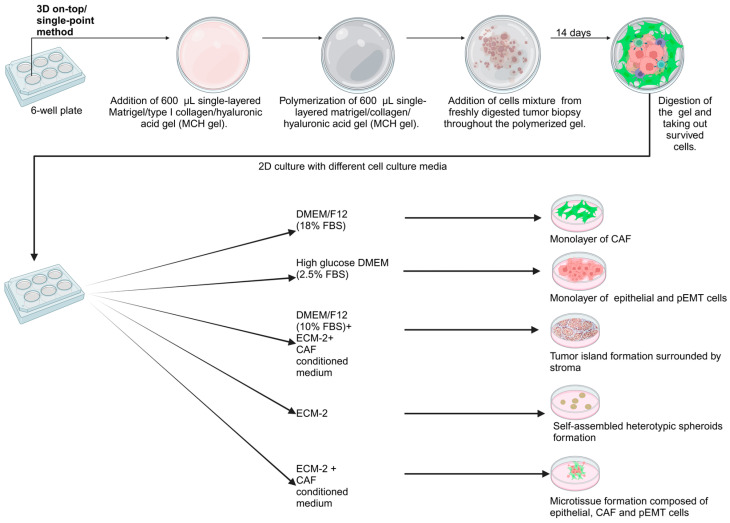
Primary culture of patient-derived tumor cells suspension after proteolytic dissociation of the tumor biopsy. Cryopreserved tumor cell suspension was cultured first on top of the MCH gel. After 14 days, the cells were extracted from the gel and propagated in 2D with different cell culture media. Different cell culture media resulted in the survival of various cell types and the spontaneous formation of tissue-like tumor structures (such as “tumor islands” surrounded by stromal areas).

**Figure 2 cells-14-01543-f002:**
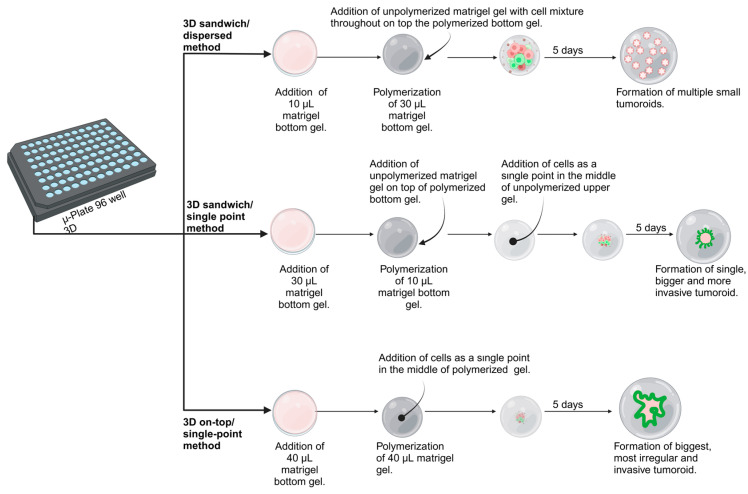
The cell seeding method influences the morphology and formation of tissue-like structures in tumoroids within 3D cultures. Three strategies were evaluated using equal cell numbers (5000 cells/well): **Top panel**: 3D sandwich/dispersed method: Cells were homogeneously embedded within a thick, unpolymerized upper gel layer, generating 3D cultures with reduced access to oxygen and nutrient diffusion. This growth condition resulted in small, slow-growing, uniformly distributed organoids with typically regular morphology. **Middle panel**: 3D sandwich/single-point method: Cell suspensions were injected into the center of a thin unpolymerized upper gel layer (but not on top), allowing improved semi-embedded formation of cells at high density, and improved access to oxygen and nutrients. This method led to the formation of a single, dense tumoroid. **Bottom panel**: 3D on-top/single-point method: The cell suspension was placed as a droplet on top of the polymerized gel surface with minimal medium, resulting in high cell density, and offering maximal oxygen and nutrient exposure. This last approach produced largest, most irregular, and rapidly growing tumoroids.

**Figure 3 cells-14-01543-f003:**
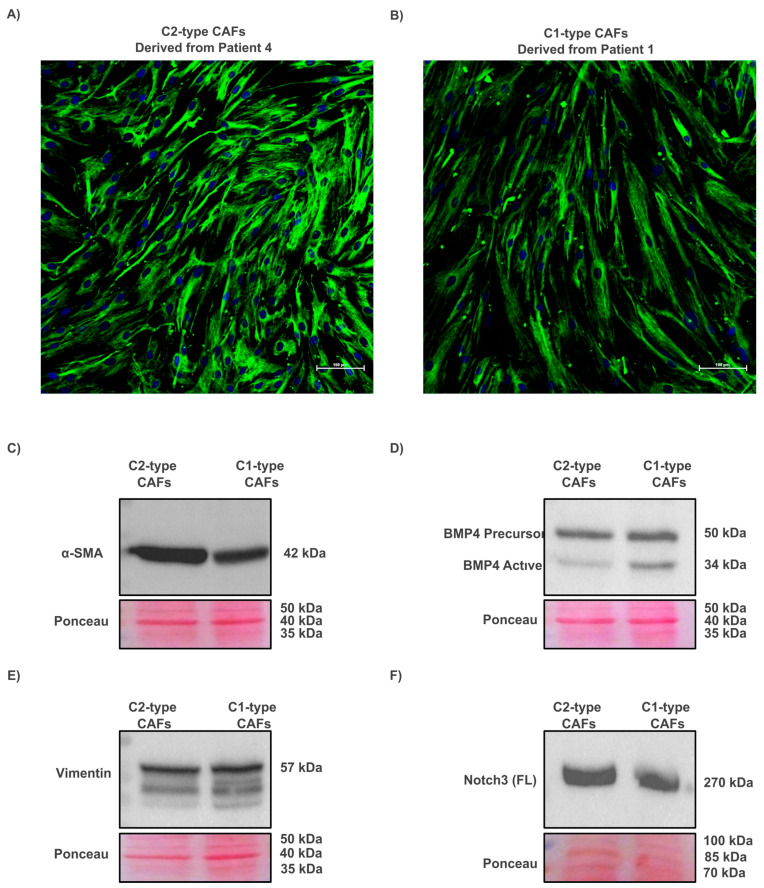
Characterization of CAF subtypes based on morphology and protein expression. (**A**) IF staining of C2-type CAFs showing strong α-SMA expression. (**B**) IF staining of C1-type CAFs with a weaker α-SMA signal. Nuclei were counterstained with Hoechst 33342. (**C**–**F**) Western blot analysis of CAFs from two patients: (**C**) α-SMA, (**D**) BMP4, (**E**) vimentin, and (**F**) full-length NOTCH3. Ponceau S staining was used as a loading and transfer control.

**Figure 4 cells-14-01543-f004:**
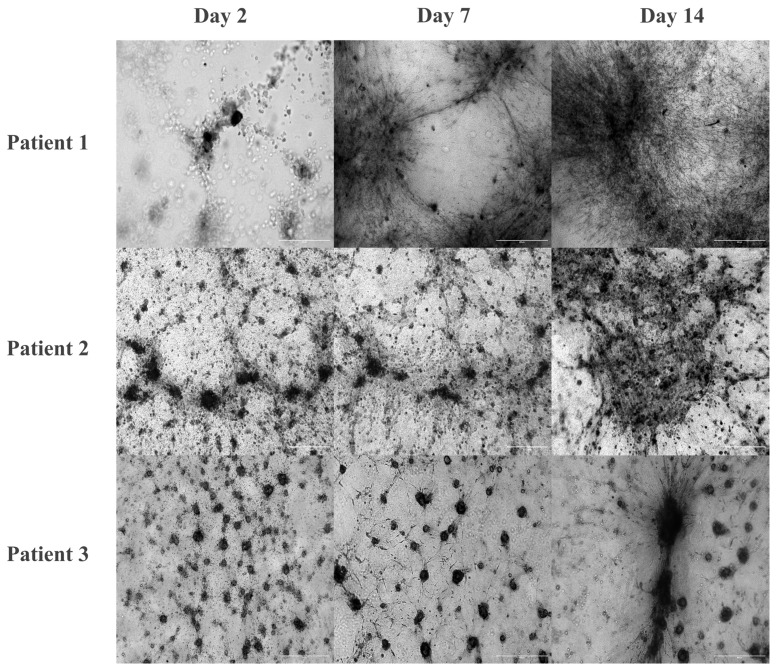
Time-dependent adaptation of tumor biopsy-derived cell suspensions on top of the MCH gel. Time-course images of tumor cell suspensions from three patients cultured on top of MCH gels at days 2, 7, and 14. Scale bar: 600 μm.

**Figure 5 cells-14-01543-f005:**
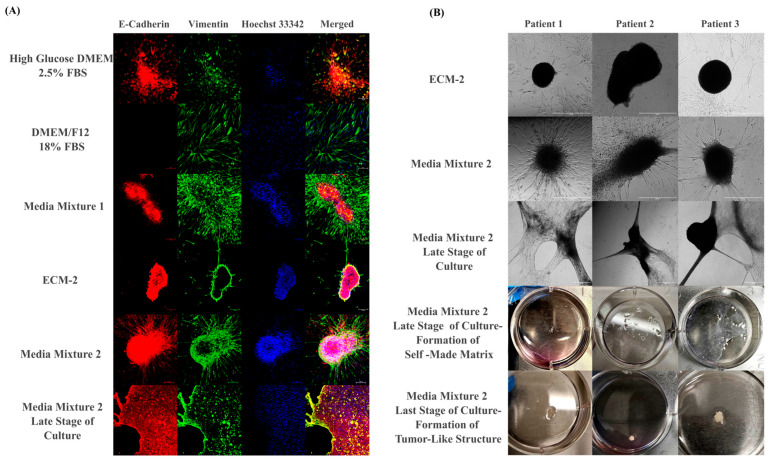
Culture media cause cell-type selection and 3D structure formation in 2D-expanded primary cultures. (**A**) IF staining of patient 1 cultures grown on plastic under different media conditions, showing E-cadherin (epithelial, red), vimentin (mesenchymal, green), and nuclear counterstaining with Hoechst 33342 (blue). Simultaneous expression of E-cadherin and vimentin indicates partial epithelial-to-mesenchymal transition (pEMT). Scale bar: 100 μm. (**B**) Brightfield and macroscopic views of cultures from patients 1–3 under spontaneous aggregation conditions. ECM-2 promoted the formation of nonadherent spheroid-like tumor aggregates, while Media Mix 2 supported adherent, invasive 3D structures. Scale bar: 600 μm.

**Figure 6 cells-14-01543-f006:**
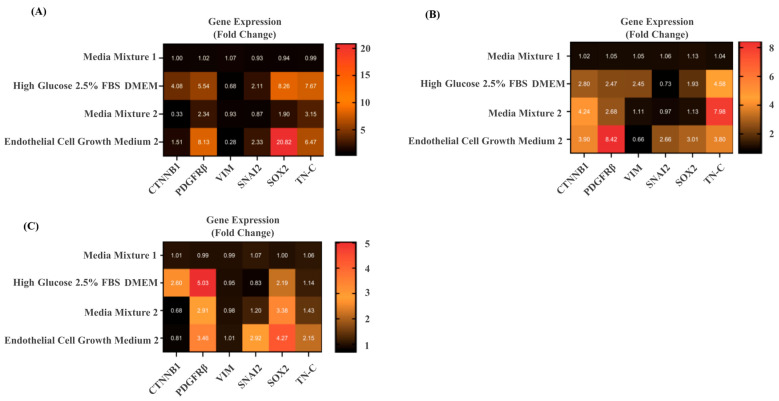
Different media have a profound effect on the expression of marker genes related to partial EMT and/or indicating stem cell characteristics. (**A**) Relative gene expression for patient 1 cell cultures in 2D, exposed to different media. (**B**) Relative gene expression for patient 2 exposed to different media. (**C**) Relative gene expression for patient 3 using the same media variations. Fold change values under 1 were considered as downregulated (black, *p* ≤ 0.05) and ratios ranging from 1.93 to 20.82 as upregulated (red, n = 3, *p* ≤ 0.05), respectively. Values represent fold change in expression calculated by the 2^−ΔΔCt method, normalized to GAPDH and expressed relative to Media Mixture 1 (calibrator = 1.0).

**Figure 7 cells-14-01543-f007:**
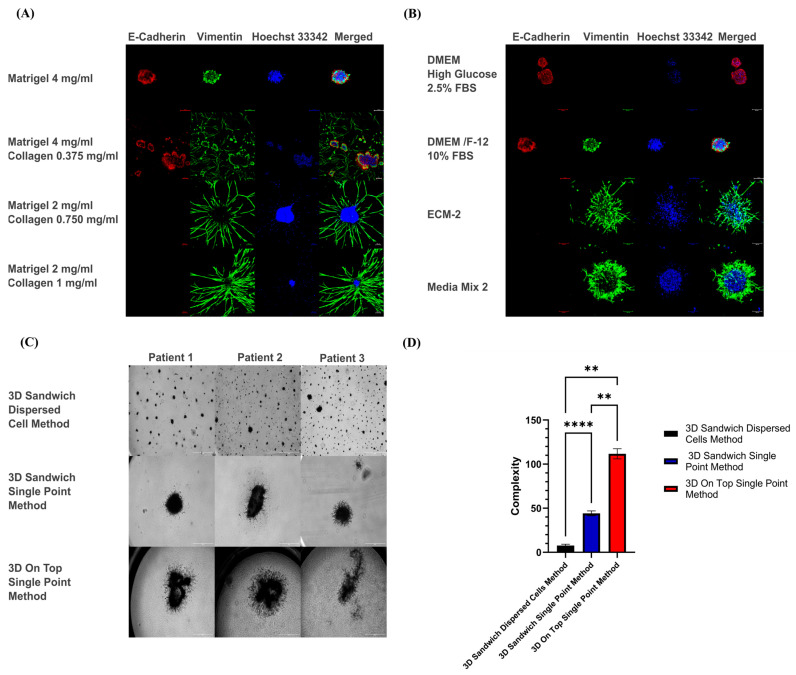
The gel composition, media, and seeding method have a profound effect on the morphology of organoids forming in 3D cultures. (**A**) Different gel composition affects the morphology of patient 1 cultures, which were treated with DMEM/F12 10% FBS and seeded with the 3D sandwich/single-point method. Scale bar: 100 μm. (**B**) Phenotypic effects of tumoroids forming from patient 1 primary cell isolates on Matrigel as a scaffold, and different media as described above. Cells were seeded using the 3D sandwich/single point method on top of a single, uniform gel layer. Scale bar: 100 μm. (**C**) The effects of different seeding methods on the morphology of patients 1, 2, and 3: All cultures contain Matrigel as a consistent hydrogel and use Media Mix 2 as media. The scale bar: 600 μm. (**D**) Complexity comparison for different cell-seeding methods. The complexity is shown as the mean ± SD (** *p* < 0.01, **** *p* < 0.0001; n = 3, one-way ANOVA).

**Figure 8 cells-14-01543-f008:**
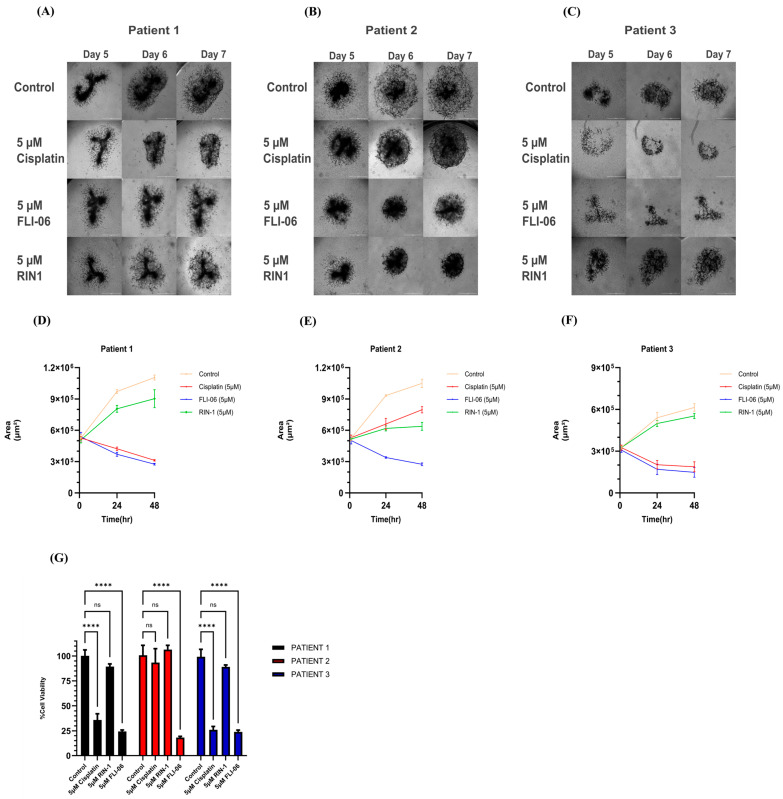
Different compounds differentially affect tumoroid growth and viability. (**A**) After 5 days of seeding, tumoroids of patient 1 cells in media mixture 2 were exposed to 5 µM cisplatin, 5 µM FLI-06, and 5 µM RIN-1. The morphologic changes were observed on days 5, 6, and 7. (**B**,**C**) Same in vitro drug sensitivity test performed with cells from patients 2 and 3, respectively. (**D**) Total area changes as observed after exposing patient 1’s cultures to different drugs. (**E**,**F**) Same in total area change for patients 2 and 3, respectively. (**G**) WST-8 metabolic assay for patients 1, 2, and 3 cultures that were exposed to cisplatin, FLI-06, and RIN-1 at day 7. Data are presented as mean ± SD (**** *p* < 0.0001, ns: not statistically significant; n = 3, two-way ANOVA with multiple comparisons).

**Table 1 cells-14-01543-t001:** Clinicopathological characteristics of four patients with oral and base of tongue malignancies. Abbreviations: LN, lymph node; pTNM, pathological tumor-node-metastasis stage; Pre-op, pre-operative.

N° Patients	Age	Sex	Site	Histology	Size (cm)	LN Status	pTNM	Stage	Grade	Pre-op Tx
**1**	72	Female	Right oral tongue	Keratinizing squamous cell carcinoma	3.5 × 4.0 × 1.5 cm	2/2 level III (+), 16 nodes (−), others reactive	pT3N2M0	IVa	G1/G2	None
**2**	41	Male	Right mandible and floor of the mouth	Keratinizing squamous cell carcinoma	5.5 × 3.5 × 5.0 cm	7/19 total positive	pT4aN2aM1	IVc	G2–G3	None
**3**	63	Female	Base of tongue	Adenoid cystic carcinoma	2.8 × 2.0 × 1.5 cm	0/7 all negative	pT2N0M0	II	Intermediate grade (cribriform + tubular), minor solid areas	None
**4**	62	Male	Left oral tongue and floor of the mouth	Keratinizing squamous cell carcinoma (recurrent)	6.5 × 4.0 × 2.0 cm	0/5 all negative	pT4aN0M0	IVa	G2	None
**N° Patients**	**Surgical–Pathologic Findings**					**Recurrence/Metastasis**				
**1**	Angioinvasion present; 2/2 LN level III positive; 0.1 cm deep margin					No local recurrence or distant metastasis				
**2**	Perineural and angioinvasion; narrow margin; 7 LN positive (1/1, 4/4, 2/14)					Extensive lung metastases, local recurrence				
**3**	Infiltrative growth into tongue, pharynx, tonsil; nerve and vessel resection; perineural invasion, Ki67 ~17%					No current evidence of distant metastasis				
**4**	Deep ulcerated mass; nodal necrosis; clear margins; R1 margin (deep)					Recurrent tumor, but no distant metastasis described				

## Data Availability

The data presented in this study are available in the presented article and Appendix A. Any additional data related to this study is available on request from the corresponding author.

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
