# Peer review of "A Patient-Derived Scaffold-Based 3D Culture Platform for Head and Neck Cancer: Preserving Tumor Heterogeneity for Personalized Drug Testing"

_cells, 2025, doi:10.3390/cells14191543_

Round 1

Reviewer 1 Report

Comments and Suggestions for Authors

Major comments:

The abstract lacks some information for context and some abbreviation that are not explained.

Figure 3: why are only data from two patients provided? For WB: a quantification and comparison across all samples would help to understand the differences between the culture conditions and the 4 different patients.

Figure 4, 6 and 7: Why are the data from patient 4 not shown?

Figure 6: it is unclear how the fold change in gene expression was calculated, please elaborate.

Single point vs dispersed cell method: the authors claim that the main difference is the availability of oxygen and nutrients. Please elaborate on the possibility that cell-cell contacts might play a role in the different phenotypes as well.

Discussion: the authors did not elaborate on the importance of the immune cells in the TME. A discussion how this important cell type of the TME could be incorporated into the in vitro system should be provided.

Minor comments:

In the introduction there is an incomplete sentence (line82)

Line 463: “After selecting Matrigel as the hydrogel of choice”: the study did not evaluate alternative hydrogels. Please rephrase.

Author Response

Reviewer 1

#Comment 1: The abstract lacks some information for context and some abbreviations that are not explained.

Response: We revised the abstract section by verifying the accuracy of abbreviations and ensuring consistency of context. Specifically, we expanded the Abstract to provide a clearer clinical and methodological context for HNSCC heterogeneity and patient-specific modeling. Furthermore, we also defined all abbreviations at first mention (e.g., WST‑8, ECM‑2, CAFs, pEMT, PDOs/PDX) and clarified what was measured (morphology/complexity and viability).

#Comment 2: Why are only data from two patients provided? For WB: a quantification and comparison across all samples would help to understand the differences between the culture conditions and the four different patients.

Response: We clarified that, among four CAF cultures, two representative subtypes were selected based on differential cell morphology and growth rate (C1 cells grow slowly/elongated cell phenotype; C2 cells grow much faster/less elongated cell morphology), which is consistent with the results by Patel et al. [5]. These two representative cultures were then subjected to immunostaining (IF) and Western blotting (WB) to illustrate subtype-specific protein expression differences. We aimed to highlight the contrast between CAF subtypes rather than providing a quantitative comparison across all four patients. We added and explained this rationale in more detail in “Results” and referenced it appropriately in the legend of Fig. 3.  The text added to the revised manuscript is as follows: “Out of four patient-derived CAF cultures, two representative subtypes were selected based on cell morphology observed by light microscopy and proliferation rate [5]. The rapidly proliferating, less elongated C2-type CAFs exhibited strong α-smooth muscle actin (α-SMA) staining, whereas the elongated, slower-growing C1-type CAFs showed weaker α-SMA signals and lower cell density. Western blotting showed subtype-specific protein expression. C2-type CAFs exhibited higher levels of α-SMA and NOTCH3 proteins, but lower levels of BMP4, whereas C1-type CAFs showed lower levels of α-SMA and NOTCH3, but higher BMP4 expression. Vimentin expression was comparable between the two subtypes. Representative IF and Western blot images are shown in Fig. 3, with full uncropped blots provided in Supplementary Fig. 1.” Where: Results section; Fig. 3 legend.

#Comment 3: Figure 4, 6, and 7: Why are the data from patient 4 not shown?

We apologize if the issue with patient 4 has caused confusion among both reviewers, and we have attempted to clarify this as clearly as possible here. The same problem with the cells of patient #4 has also been noted by reviewer 2 (connected comments #9, #18, #19, and #21). We try not to unnecessarily repeat our detailed answer to this issue at several places in this letter. Therefore, we provide a thorough, primary response here that hopefully answers all other occasions where the patient 4 issue is discussed.

Response: We now explained in more detail that Patient 4 CAFs exhibited a C2-type hyperproliferative phenotype, and we could not use patient #4 cell isolates for our drug testing due to these properties.  For this reason, patient #4 cells (and only the CAFs from patient #4) were only included for CAF characterization and conditioned medium preparation (Fig. 3), but they were not suitable for our tumoroid drug testing or drug‑response assays. This is now stated explicitly and hopefully more clearly in the text, where the description of those experiments begins. For drug response assays (Figs. 4, 6, and 7), we focused entirely on patients 1–3 to maintain a consistent dataset across all experiments and to demonstrate the proof of concept. We have also clarified this point in the Results section. This section was added for clarification: “Because previously CAFs from Patient 4 displayed a C2-type and a strong hyperproliferative phenotype, these cells were therefore used only for CAF medium preparation and stromal characterization (Fig. 3), and all the following experiments were conducted with Patients 1–3 only to maintain consistency across assays. “ Where: Results (before Fig. 4) and relevant figure legends.

#Comment 4: Figure 6: It is unclear how the fold change in gene expression was calculated. Please elaborate.

Response: We thank the reviewer for this comment. Fold changes shown in the heatmaps were calculated using the 2^−ΔΔCt method. Ct values were first normalized to the house-keeping gene GAPDH (ΔCt). For each gene, expression observed in “Media Mixture 1” was used as the calibrator (set to 1.0), and fold change was calculated relative to this condition. We have now clarified this in the Methods section and the figure legend. We added a short sentence to explain our procedure: “Values represent fold change in expression calculated by the 2^−ΔΔCt method, normalized to GAPDH and expressed relative to Media Mixture 1 (calibrator = 1.0).” This is also commented by reviewer 2, comment #20. In addition, we have increased the font size of axis labels, numbers, and the color scale to enhance readability. Where: Methods (qRT‑PCR subsection); Fig. 6 legend.

#Comment 5: Single point vs dispersed cell method: The authors claim that the main difference is the availability of oxygen and nutrients. Please elaborate on the possibility that cell-cell contacts might play a role in the different phenotypes as well.

Response: We certainly agree that altered cell–cell contacts may also contribute to the phenotypic differences between single-point and dispersed seeding. We added text noting that increased local cell density in our single-point seeding technique likely enhances cell–cell interactions, which, together with oxygen/nutrient diffusion, contribute to the more complex tumoroid phenotypes. We also mirrored this point in the Discussion. We added a sentence just before figure 7: “In addition to nutrient and oxygen gradients, the higher local cell density in single-point cultures may also increase cell–cell contacts, which could contribute to the more complex tumoroid phenotypes observed.”  Where: Results (seeding‑topology paragraph); Discussion.

#Comment 6: Discussion: The authors did not elaborate on the importance of the immune cells in the TME. A discussion on how this important cell type of the TME could be incorporated into the in vitro system should be provided.

Response: We agree that immune cells are a key component of TME, with a significant impact on tumor behavior and therapy response. However, we also state that we are currently unable to stabilize/maintain their presence in our cultures for more extended periods. This was also not the primary focus of our investigations, which primarily focused on the interaction between tumor cells and CAFs. In line with the reviewer’s suggestion, we added a Limitations/Future Directions paragraph acknowledging the absence of immune components in the current experimental system and outlining feasible next steps (e.g., integration of autologous or peripheral‑blood-derived immune cells for co-culture and immunomodulatory testing), with appropriate citations. We added a detailed section to elaborate on these issues in the discussion: “Another significant limitation of our current system is the absence of immune cells, which are central components of the tumor microenvironment (TME). Immune cells, including tumor-associated macrophages, T cells, and myeloid-derived suppressor cells, play a critical role in influencing tumor progression and therapeutic response [51]. Their integration into 3D culture systems has the potential to enhance physiological relevance and allow investigation of immunomodulatory therapies. Future studies could address this by incorporating autologous immune cells or peripheral blood–derived immune populations into the co-culture system. The absence of histopathological sectioning and direct comparison of tumoroids with matched patient tumors; future studies will incorporate histological evaluation by pathologists to further validation“.  Where: Discussion (Limitations/Future Directions).

#Comment 7: In the introduction there is an incomplete sentence (line82)

Response: Corrected; the relevant sentence is now complete and grammatically correct.

#Comment 8: Line 463: “After selecting Matrigel as the hydrogel of choice”: the study did not evaluate alternative hydrogels. Please rephrase.

Response: We did not screen multiple hydrogel types, but instead compared Matrigel alone with its combination with type I collagen. The sentence has now been rephrased to a more neutral description of choosing Matrigel for stability/CAF support, without implying prior multi-hydrogel screening (which did not occur here).  It now sounds like this: “After evaluating Matrigel alone and in combination with type I collagen, Matrigel was selected as the hydrogel for subsequent experiments. Where: Results (optimization of 3D growth conditions).

Reviewer 2 Report

Comments and Suggestions for Authors

A Patient-Derived Scaffold-Based 3D Culture Platform for Head and Neck Cancer: Preserving Tumor Heterogeneity for Personalized Drug Testing, by Anameriç et al.

In this study, the authors present a method on how to grow 3D cultures (PDOs) from patient-derived head-and-neck cancer with the aim of using them for drug testing. The authors test different tissue hydrogels/scaffold for their ability to support ex vivo regrowth/adaptation, optimize cell culture media for 3D culture propagation and development, and test response to cisplatin and 2 Notch modulators (RIN and FLI-06) in 3D cultures. The authors conclude that they have “developed a robust and reproducible scaffold-based 3D culture platform for personalized medicine in head and neck cancer”, which can be used for “for guiding individualized therapeutic decisions”.

The observations are interesting and an important initiative to develop procedures that allows optimization of therapeutic strategies for head and neck cancer patients. Overall, the paper, however, appears as a preliminary draft that needs extensive improvement. It is an over-interpretation to claim to have “developed a robust and reproducible platform” based on 3 patient samples.

I have collected some comments in the following:

It is not clear why patient #4 is excluded from most figures. Although that patient #4 was not available for drug testing should not prevent its use for the other analyses.

Text fragments appear incomplete: line 82, 398

2.4: Please clearly specify what is the content of the MCH scaffold - It seem not to be specified. Lines 173-183 do not belong under the subtitle “Preparation of hydrogels”.

2.6: It seem evident that tht in Fig 5, images were obtained  from cell cultures grown on plastic. It is assumed that cryo or paraffin sections that were prepared for Fig S2? How were the organoids prepared for these analyses? 

2.8: Line 263-264: isn’t HA part of the gel composition?

  1. 1: Line 331: It is recommended that the authors state what they observe, and not use this indirect type phrasing: “Western blotting confirmed subtype-specific protein expression: C2-type CAFs expressed high α-SMA and NOTCH3 but low BMP4, while C1-type CAFs showed the opposite profile.”

3.3: The authors refer to “histology” analyses of tumor islands and stroma. However, the manuscript does not contain any histology data of the organoids (only macroscopic/morphology data). The paper would benefit strongly from evaluation of the organoid structures by histological examination. E.g. by direct comparison performed by a pathologist’s histological description of the primary tumor and the organoid. This would allow better conclusion on “which media keep the original tumor heterogeneity best”.

The Discussion section is long. The authors should shorten the Discussion by focusing on the interpretation of the key findings. The Discussion is presenting background information (e.g. 511-526) without elaborating on the findings. Repeating results (e.g. 553-574) without adding to the interpretation. The Discussion also brings the reader into situations, where it is unclear if the observations are done by the authors or by others (e.g. 561-564).

Table 1. Abbreviations (LN, pTNM and Pre-op) used here should be specified in the legend.

Fig 4. Why is patient 4 excluded. There is only one Panel – therefore omit A).

Fig 5. Why is patient 4 excluded from B? Bar annotations can hardly be seen.

Fig 6. The qPCR data are not easily interpreted. It seems that all ”Media Mixture 1” are black, but they seem to be the reference parameter? Specify in the legend what are the fold change values related to, or how the numbers are generated. Font is very small.

Fig 7. Why is patient 4 excluded from C?

Fig 8. It is possible that the effect of cisplatin seen in the images is linked to the Area measures and the viability. This is not interpreted from those treated with FLI-06 in patient 1. Also, the viability after FLI-06 (G) seem not to be linked with the observation on the 3D structure (A and C). Font is very small.

Suppl Fig 2. This figure can hardly be interpreted. Epithelial markers vary, but so do also the organoid (condition?). The upper line may represent the same organoid with Vimentin and SMA, but the other lines are obviously not the same as judged from structure. DAPI staining suggest that the staining is done on sections. How were the sections prepared?

Author Response

Reviewer 2

#Comment 9: It is not clear why patient #4 is excluded from most figures. Although that patient #4 was not available for drug testing should not prevent its use for the other analyses.

Response: We have already addressed the same issues in response to Comment #3 of Reviewer 1, where the question is discussed in detail. It is evident that this section lacked clarity and required detailed explanations and needed fixing. Please check our response above.

#Comment 10: Text fragments appear incomplete: line 82, 398

Response: Again, we have already addressed these issues in our response to reviewer 1, who identified the same problems.  We corrected the incomplete sentences, rewrote/merged fragmented lines; methods and figure legend text now read as complete, coherent sentences.

#Comment 11: 2.4: Please clearly specify what is the content of the MCH scaffold - It seem not to be specified. Lines 173-183 do not belong under the subtitle “Preparation of hydrogels”.

Response: We have now specified the exact composition and preparation of the Matrigel/Collagen/Hyaluronic Acid (MCH) hydrogel (2 mg/mL Matrigel, 1 mg/mL collagen type I, and 2% HA w/v) at first mention in the Methods. In addition, we refined the Methods structure by introducing clear subheadings. To address this issue, we created a dedicated section “Preparation of Matrigel/Collagen/Hyaluronic Acid (MCH) hydrogel” subsection with the exact final concentrations (Matrigel, collagen I, HA). We clarified that HEPES was used for buffering. The new paragraph reads as follows: “Matrigel (phenol-red free, Matrigel®; Corning, Germany), type I collagen (rat tail collagen type I; Corning), HEPES buffer (1 M; Gibco™, Thermo Fisher Scientific), and hyaluronic acid (HA) were used to prepare the MCH hydrogel. Hyaluronic acid was prepared as described previously with minor modifications [19]: Briefly, 50 mg of Hyaluronic acid sodium salt (from Streptococcus, Thermo Fisher Scientific) was stirred with 5 ml of PBS containing Ca2+ and Mg2+ ions overnight. The pH was measured and adjusted to neutral pH 7.0. Gels were finally prepared at a final concentration of 2mg/mL Matrigel, 1mg/mL type I collagen, and 2%HA (w/v).” These changes clarify the workflow and ensure that the methodological descriptions are correctly placed under the appropriate sections. Where: Methods (new MCH hydrogel subsection).

# Comment 12: It seem evident that tht in Fig 5, images were obtained  from cell cultures grown on plastic. It is assumed that cryo or paraffin sections that were prepared for Fig S2? How were the organoids prepared for these analyses?

Response:  The structures shown in Supplementary Figure 2 are non-adherent aggregates or organoids that are spontaneously detached from the plates in ECM-2 cultures. After detachment, these free-floating structures were collected, centrifuged, fixed in 4% PFA, and directly processed for immunofluorescence staining & confocal microscopy, using an optical clearing protocol [reference # 20], without embedding or sectioning. To clarify this point, we added detailed description for all handling steps (detachment/collection, centrifugation, 4% PFA fixation, optical‑clearing, and whole‑mount staining of organoids): “Media mix 1 (which retained 33% growth-factor-enriched ECM-2) effectively promoted the spontaneous formation of squamous carcinoma-like tissue-like morphology, characterized by tumor island–like structures surrounded by stromal zones. ECM-2 medium alone generated non-adherent spheroid-like aggregates consisting of tumor and pEMT cells. Still, it failed to retain CAFs, as verified by immunostaining of different aggregates for stromal and epithelial markers (Supplementary Fig. 2). “ Where: Methods (IF staining for 2D cultures; note on aggregates from ECM‑2 or late Response.

# Comment 13: 2.8: Line 263-264: isn’t HA part of the gel composition?

Response: We clarified that HA is indeed part of the initial MCH gel used for direct 3D culture from biopsy; only for µâ€‘plate drug‑testing assays, we used pure Matrigel (or Matrigel/collagen mixes where applicable). Hyaluronic acid (HA) was included in the initial hydrogel preparations (MCH) described earlier; however, HA was not used in the subsequent µ-Plate drug testing platform. For these experiments, we focused on Matrigel alone and Matrigel/Collagen I mixtures to establish reproducible and stable conditions. The following paragraph was added: “Matrigel (phenol-red free, Matrigel®; Corning, Germany), type I collagen (rat tail collagen type I; Corning), HEPES buffer (1 M; Gibco™, Thermo Fisher Scientific), and hyaluronic acid (HA) were used to prepare the MCH hydrogel. Hyaluronic acid was prepared as described previously with minor modifications [19]: Briefly, 50 mg of Hyaluronic acid sodium salt (from Streptococcus, Thermo Fisher Scientific) was stirred with 5 ml of PBS containing Ca2+ and Mg2+ ions overnight. The pH was measured and adjusted to neutral pH 7.0. Gels were finally prepared at a final concentration of 2mg/mL Matrigel, 1mg/mL type I collagen, and 2%HA (w/v). “ Where: Methods (MCH section and µâ€‘plate assay description).

# Comment 14: 3.1: Line 331: It is recommended that the authors state what they observe, and not use this indirect type phrasing: “Western blotting confirmed subtype-specific protein expression: C2-type CAFs expressed high α-SMA and NOTCH3 but low BMP4, while C1-type CAFs showed the opposite profile.”

Response 14: We have revised the sentence to “Western blotting showed subtype-specific protein expression …” to directly describe our observations rather than using indirect phrasing.

# Comment 15: 3.3: The authors refer to “histology” analyses of tumor islands and stroma. However, the manuscript does not contain any histology data of the organoids (only macroscopic/morphology data). The paper would benefit strongly from evaluation of the organoid structures by histological examination. E.g. by direct comparison performed by a pathologist’s histological description of the primary tumor and the organoid. This would allow better conclusion on “which media keep the original tumor heterogeneity best”.

Response: We agree that histopathological sectioning and pathologist-based evaluation would further strengthen the study. However, due to hospital policy and ethical restrictions, histological embedding of patient-derived cultures was not feasible within the scope of this project. Instead, we characterized the cultures using morphological assessment, complexity scoring, and immunofluorescence analyses, which allowed us to identify tumor island–like structures surrounded by stromal zones. To avoid overstatement, we have revised the text to replace “histology” with “tissue-like morphology/architecture.” We also added a statement in Study Limitations and Future Directions acknowledging that future studies should incorporate histological evaluation and direct comparison with matched patient tumors to more rigorously determine which media best preserve tumor heterogeneity. Where: Results (media‑effects paragraph); Discussion (Limitations).

# Comment 16: The Discussion section is long. The authors should shorten the Discussion by focusing on the interpretation of the key findings. The Discussion presents background information (e.g. 511-526) without elaborating on the findings. Repeating results (e.g. 553-574) without adding to the interpretation. The Discussion also brings the reader into situations, where it is unclear if the observations are done by the authors or by others (e.g. 561-564).

Response: We agree with the reviewer that the discussion can be more concise. We have now streamlined the background information, consolidated overlapping text, and refocused on the three key parameters (scaffold, media, seeding) with matching interpretations and citations. We also added a concise Limitations/Future Directions section at the very end. We believe this version is more straightforward, focused, and aligned with the reviewer’s recommendation. We have attached the new discussion at the end of this response letter.

# Comment 17: Table 1. Abbreviations (LN, pTNM, and Pre-op) used here should be specified in the legend

Response 17: We have revised the legend of Table 1 to include definitions of the abbreviations (LN, pTNM, and Pre-op) for clarity. Where: Table 1 legend.

# Comment 18 : Fig 4. Why is patient 4 excluded. There is only one Panel – therefore omit A).

Response: Fig. 4 is now labeled as a single-panel figure without extraneous sublabels. As explained in our response to comment #2 of reviewer 1, patient 4 CAFs displayed a C2-like, hyperproliferative cellular phenotype, which made them unsuitable for tumoroid drug testing and downstream functional assays. Therefore, Patient 4 was not included in Figs. 4–7, to maintain a consistent dataset across patients. Where: Fig. 4 legend.

# Comment 19: Fig 5. Why is patient 4 excluded from B? Bar annotations can hardly be seen

Response: Fig. 5 legend now specifies “patients 1–3,” consistent with the text; Patient 4 rationale is provided as outlined above. As noted earlier, Patient 4 was excluded from functional assays due to its C2-type hyperproliferative CAF phenotype. This has now been explicitly clarified in the Results and is outlined in detail in our response to comment #3 of reviewer 1.  Where: Fig. 5 legend.

# Comment 20: Fig 6. The qPCR data are not easily interpreted. It seems that all ”Media Mixture 1” are black, but they seem to be the reference parameter? Specify in the legend what are the fold change values related to, or how the numbers are generated. Font is very small.

Response: This was also commented on by reviewer 1 (comment # 4) and has been answered in detail on this occasion. In short, we added an explicit 2^−ΔΔCt definition and referenced the calibrator, which we believe improved labeling clarity. Additionally, we have increased the font size of axis labels, numbers, and the color scale to enhance readability. Where: Fig. 6 legend; Methods

# Comment 21: Fig 7. Why is patient 4 excluded from C?

Response: Patient 4 was excluded from Fig. 7C for the same reason as in all other drug testing–related experiments. Only the CAFs from patient #4  were used for CAF characterization and conditioned medium preparation. Please review our detailed response to this issue, which addresses comment #3 from reviewer 1.  Where: Results (before Fig. 4) and relevant figure legends.

# Comment 22: Fig 8. It is possible that the effect of cisplatin seen in the images is linked to the Area measures and the viability. This is not interpreted from those treated with FLI-06 in patient 1. Also, the viability after FLI-06 (G) seem not to be linked with the observation on the 3D structure (A and C). Font is very small.

Response: We thank the reviewer for bringing this to our attention. We now clarified that detached halos of dead/dying cells visible in brightfield after FLI‑06 were excluded from area measurements. These areas were not included in the ImageJ area quantification, which was limited to dense, viable regions. Specifically, the blurred appearance of some FLI-06–treated tumoroids in Fig. 8 reflects shadows from detached/dead cells, rather than indicating preserved tumor growth, as shown in the close-up below. The results, therefore, show a consistent reduction in both area and viability after FLI-06 treatment. In parallel, results from simultaneous WST-8 and morphologic analyses indicated reduced viability/size across all three patients. We have clarified this point in the Results section. Where: Results (drug‑testing paragraph) and Fig. 8 legend.

Fig 8A FLI-06 treatment close-up with higher resolution.

# Comment 23: Suppl Fig 2. This figure can hardly be interpreted. Epithelial markers vary, but so do the organoids (conditions?). The upper line may represent the same organoid with Vimentin and SMA, but the other lines are obviously not the same as judged from structure. DAPI staining suggest that the staining is done on sections. How were the sections prepared?

Response: To address this question, please also refer to our response to Comment # 12; the methods section was updated and now provides more detail on the preparation and staining of non-adherent aggregates used in the supplemental figure. The following sentences were added for clarification: “For non-adherent spheroid-like aggregates formed in ECM-2 cultures and tumor-like structures arising during late-stage Media Mix 2 cultures, detached material was collected from the medium, centrifuged, and fixed in 4% PFA. These structures were subsequently processed and stained using an optical clearing protocol for 3D cultures [20], without embedding or sectioning. “

To summarize, Supplementary Fig. 2 shows six representative, non-adherent aggregates derived from patient 1 that were generated (or spontaneously formed) under identical cell culture conditions using growth-factor-enriched ECM-2 media. The figure shows different aggregates rather than serial stains/sections of a single aggregate, which explains minor shape differences. No sectioning was performed; aggregates were collected after spontaneous detachment, fixed, and imaged as intact whole-mounts using an optical clearing protocol [20]. The aggregates shown in Fig. S2 were stained with different epithelial (E-cadherin, pan-cytokeratin, β-catenin) and stromal (vimentin, α-SMA) markers to demonstrate that only tumor and pEMT cells are present inside the aggregates. At the same time, CAFs were excluded or were growing exclusively attached to the plastic surface. Where: Methods (IF staining; aggregate handling note).

Round 2

Reviewer 2 Report

Comments and Suggestions for Authors

To help the reader, the results/observations related to the five different growth media, should be summarized in a Table, where columns could be “promote epithelial growth”, “promote CAFs”, “promote invasive morphology” - the authors should consider including the most relevant.

Text font in Fig 5 and 6 are still critically small.

There is quite some literature on Notch signaling and cisplatin resistance, which should be touched in the last part of the Discussion, and not cut by "warrants further study".

Author Response

Answers to Reviewer 2 – Reviewing Round #2

Comment 1: To help the reader, the results/observations related to the five different growth media, should be summarized in a Table, where columns could be “promote epithelial growth”, “promote CAFs”, “promote invasive morphology” - the authors should consider including the most relevant.

Answer 1: Thanks for the suggestion, we have generated such a table (see below), but then we  considered skipping this option, as larger tables can also distract from the main text flow. We could consider asking the editors opinion about this and then get it implemented, or not. In addition, we have aimed at a clear description of the results in the text and the respective section should now be clearly structured by adding paragraphs and subsections, one for each of the main observations. We honestly think this should be clear as it is now.

Comment 2: Text font in Fig 5 and 6 are still critically small.

Answer 2: The font sizes in both figures have been increased and should now be more readily legible when the figures are embedded in the manuscript.

Comment 3: There is quite some literature on Notch signaling and cisplatin resistance, which should be touched in the last part of the Discussion, and not cut by "warrants further study".

Answer 3: we agree with the reviewer that this would improve the manuscript, and we have now added a short paragraph to the end of the discussion: “As shown in the clinics and various models, increased NOTCH1 signaling/activity often correlates with reduced cisplatin sensitivity. Experimental Notch blockade has been demonstrated to re-sensitize cells to cisplatin. Mechanistically, Notch sustains cancer stem-like populations and EMT/invasion programs (e.g., via the NOTCH1/NOTCH3–JAG1 axes), while also rewiring the DNA-damage response and pro-survival signaling pathways (PI3K/AKT, STAT3, NF-κB), thereby blunting cisplatin-induced apoptosis [50]. In this framework, our finding that the NOTCH modulator RIN-1 selectively restricted radial spread/area in the cisplatin-resistant tumoroids without loss of viability is consistent with other reports showing that Notch perturbation enforced a cytostatic/anti-invasive state, a phenotype also seen when Notch output is modulated in other squamous models. Prior studies have demonstrated that RIN-1 targets RBPJ and induces activation-like increases in Notch-responsive transcripts (e.g., HES1/HEY1, NOTCH3, JAG1) in specific HNSCC settings [8,51]. Furthermore, disrupting RBPJ can de-repress specific Notch targets or indirectly enhance Notch tone via ligand changes, resulting in a cytostatic, differentiation-inducing program (increased expression of HES/HEY-linked p21 CDKN1A, reduced proliferation and invasion) rather than promoting apoptosis. We propose that cisplatin-resistant tumoroids retain a Notch-responsive, stem cell- and/or EMT–enriched state. We observe that RIN-1 likely perturbs this circuitry, thus effectively reducing radial spread without an apparent loss of viability.“ This section now also cites a few relevant references specifically describing the connection between cisplatin resistance and NOTCH signaling.

TABLE 2

Table 2. Summary of media effects on epithelial vs. CAF growth and invasive morphology in scaffold-based 3D cultures

Our qualitative scores summarize observations described in Section 3.5 (with supporting details from Methods Section 2.6 and Results Sections 3.3–3.4) of the manuscript. “Promote invasive morphology” indicates the spontaneous formation of irregular, radially spreading tumoroids described in our complexity analysis.

Medium

Promote epithelial growth

Promote CAFs

Promotes invasive morphology

Key notes / context

DMEM (high glucose) + 2.5% FBS

Yes (tumor/pEMT survive)

No (does not support CAF growth)

Low (tumoroids, mainly tumor-only)

In 3D (e.g., Matrigel; sandwich/single-point seeding), yielded tumoroids predominantly composed of tumor cells; CAFs not supported.

DMEM/F12 + 18% FBS

Moderate (tends to differentiate tumor cells)

High (expands CAFs)

Low–Moderate

High-FBS conditions favor CAF outgrowth (useful for CAF isolation/expansion). Not highlighted for invasion.

Media Mix 1 (ECM-2 : CAF-CM : DMEM/F12+10% FBS = 1:1:1)

High (squamous “tumor islands”)

Moderate (stroma present)

Low–Moderate

Preserves tumor–stroma balance and organized “tumor island” morphology on plastic; less invasive than Mix 2/ECM-2.

ECM-2 (serum-reduced)

High (epithelial/pEMT enriched)

Low in 2D / High in 3D

High in 3D

Context-dependent: in 2D, enriches epithelial/pEMT while excluding CAFs (non-adherent spheroids); in 3D Matrigel, supports large heterogeneous, invasive tumoroids with active CAF contribution.

Media Mix 2 (50% ECM-2 + 50% CAF-CM)

High

High (retains/expands CAFs)

High (most invasive/complex)

In 2D, creates adherent, 3D-like invasive structures; in 3D Matrigel with single-point/on-top seeding, produces the highest morphological complexity; used for downstream drug testing.

Abbreviations: CAF, cancer-associated fibroblast; pEMT, partial epithelial–mesenchymal transition.
